# Dynamic Mixed-Prototype Model for Incremental Deepfake Detection

## ABSTRACT

The rapid advancement of deepfake technology poses significant threats to social trust. Although recent deepfake detectors have exhibited promising results on deepfakes of the same type as those present in training, their effectiveness degrades significantly on novel deepfakes crafted by unseen algorithms due to the gap in forgery patterns. Some studies have enhanced detectors by adapting to the continuously emerging deepfakes through incremental learning. Despite the progress, they overlooked the scarcity of novel samples that can easily lead to insufficient learning of forgery patterns. To mitigate this issue, we introduce the Dynamic Mixed-Prototype (DMP) model, which dynamically increases prototypes to adapt to novel deepfakes efficiently. Specifically, the DMP model adopts multiple prototypes to represent both real and fake classes, enabling learning novel patterns by expanding prototypes and jointly retaining knowledge learned in previous prototypes. Furthermore, we propose the Prototype-Guided Replay strategy and Prototype Representation Distillation loss, both of which effectively prevent forgetting learned knowledge based on the prototypical representation of samples. Our method surpasses existing incremental deepfake detectors across four datasets and exhibits superior generalizability to novel deepfakes through learning limited deepfake samples.

## CCS CONCEPTS

• **Computing methodologies** → **Biometrics**; • **Security and privacy** → **Human and societal aspects of security and privacy**.

## KEYWORDS

Deepfake Detection, Multimedia Forensics, Incremental Learning

## 1 INTRODUCTION

Recent advancements in deep generation models have facilitated the creation of highly convincing and realistic facial media, commonly known as deepfakes. The accessibility and convenience of this technology have inspired a broad range of inventive and intriguing applications, including face editing [1, 2], face swapping [3, 4], face avatar [5, 6], and talking face generation [7, 8]. Unfortunately, it can also be exploited by attackers for malicious purposes, such as telecom fraud, circumventing face recognition systems, and disseminating false information [9], thereby posing severe risks

*ACM MM, 2024, Melbourne, Australia*
© 2024 Copyright held by the owner/author(s). Publication rights licensed to ACM.
ACM ISBN 978-x-xxxx-xxxx-x/YY/MM
https://doi.org/10.1145/nnnnnnn.nnnnnnn

to public safety and causing widespread concerns. Therefore, the development of effective deepfake detection methods has become more urgent. In addition, the continued evolution and proliferation of deepfake algorithms brings challenges to the generalizability and adaptability of deepfake detectors, requiring the ability to quickly detect novel deepfake contents.

Previous works [10–12] have primarily focused on enhancing the generalizability of deepfake detectors in the absence of access to novel deepfakes. These methods improve performance on unseen deepfakes in the testing phase by guiding detectors to capture prior forgery patterns that may be shared among the training deepfakes and unseen deepfakes. Specifically, these methods incorporate strategies such as identifying face swap boundaries with synthesizing training data [10, 13, 14], spotting texture anomalies and inconsistency [11, 15], and perceiving audio-visual mismatch using pretext tasks [12, 16]. Nevertheless, different deepfake algorithms exhibit distinct forgery patterns, and learned patterns may not be universally applicable to expose various novel deepfakes. As a result, their reliance on prior patterns leads to compromised performance when applied to deepfakes that significantly deviate from these patterns.

Some studies [17, 18] resort to utilizing a limited number of samples for adaptation to detect novel deepfakes, *i.e.*, incremental deepfake detection, sidestepping the limitations of relying on prior patterns to detect unknown deepfakes. Namely, these methods assumed that a few labeled faces are available for adapting deepfake detectors, a scenario we consider plausible. For instance, multimedia retrieval [19] can be employed to retrieve the original image, thereby enabling verification of the consistency of an image with its original source to obtain its label. Additionally, proactive deepfake watermarking [20, 21] can aid in determining whether an image has been manipulated. With a small set of labeled samples, the detector can then incrementally learn to detect novel deepfakes during the inferencing phase, which is more suitable for practical application scenarios than enhancing detectors following generalizable deepfake detection methods.

Incremental deepfake detection involves multiple learning stages, each corresponding to the process of adapting the detector to the distribution of different deepfake algorithms. This requires the network to efficiently learn a few novel deepfakes while avoiding the forgetting of previously acquired deepfake patterns. Current incremental deepfake detection methods [17, 18] focus on preventing catastrophic forgetting of learned deepfakes by utilizing knowledge distillation and data replay. For instance, CoReD [17] employs the previously trained model as a teacher network and uses a distillation loss to guide the student network in the subsequent stage, thereby preventing the forgetting of learned deepfakes. Compared to CoReD, DFIL [18] further employs hard example replay to preserve the deepfake detector's knowledge of previously learned forgery patterns. These methods typically utilized softmax loss to cluster limited novel deepfakes and previous deepfakes into a

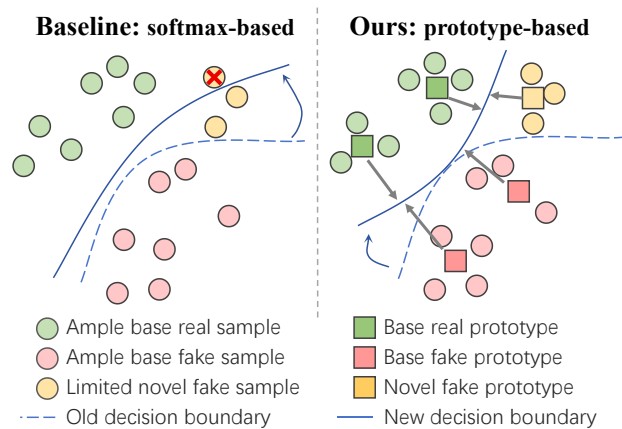

**Baseline: softmax-based**  **Ours: prototype-based**

- 🟢 Ample base real sample
- 🔴 Ample base fake sample
- 🟡 Limited novel fake sample
- ┄┄ Old decision boundary
- 🟩 Base real prototype
- 🟥 Base fake prototype
- 🟧 Novel fake prototype
- —— New decision boundary

**Figure 1: Previous approaches incrementally learn novel deepfakes by adapting detectors through `softmax` loss, whereas the scarcity of novel deepfakes often leads to insufficient learning. In contrast, the proposed DMP model explicitly learns novel deepfakes through newly added prototypes to reduce overfitting, facilitating more effective classification in the prototype space.**

single group for shared features to adapt to novel deepfake samples. However, given the variety of potential deepfake patterns, such methods hinder the learning of novel patterns. Namely, these methods do not address the insufficient learning of limited novel deepfake samples.

To adequately learn novel deepfake patterns using limited samples, we resort to prototypical networks [22, 23]. Prototypical networks classify samples in a metric space by computing distances to prototypes of each class. Compared to existing incremental deepfake detection works that cluster novel and previous deepfakes into a single group, prototypical networks reflect a simpler inductive bias that is advantageous when learning from limited samples [22]. To this end, we introduce the **Dynamic Mixed-Prototype (DMP)** model, which dynamically increases prototypes to learn novel patterns from limited incremental deepfakes efficiently, as illustrated in Figure 1. Specifically, the DMP model leverages multiple prototypes for both real and fake classes, which enables the learning of novel forgery patterns by expanding the set of fake prototypes during incremental learning. In this manner, novel deepfake patterns are learned through newly added prototypes and knowledge learned in previous prototypes is retained jointly, greatly enhancing the effectiveness of incremental deepfake detection.

During the training process on novel deepfakes, the model's ability to discriminate previously learned deepfakes can be compromised, *i.e.*, catastrophic forgetting. This arises because the learned representations are susceptible to sequential training in incremental learning, which underscores the stability of the learned representations. As the proposed DMP model represents a sample as its distance to prototypes, the stability of representations necessitates that this distance remains stable after sequential training. To address this need, we propose the **Prototype Representation Distillation (PRD)** loss, which leverages DMP to distill itself in

adapting to novel deepfakes and adopts the distance from sample to prototypes as the intermediate representations in distillation. In addition, we incorporate the **Prototype-Guided Replay (PGR)** strategy to further enhance the distillation by replaying learned deepfakes. Specifically, PGR selects several samples closest to each prototype to store in a replay set, which is replayed during the next phase of incremental learning. During distillation using PRD, PGR replays the samples closest to each prototype to maintain the stability of the learned prototypes. Adopting PRD and PGR demonstrates major advancement in maintaining learned representations and thus enhances the proposed DMP. Consisting of the DMP model, the PGR strategy, and the PRD loss, our method achieves the dual goals of mitigating catastrophic forgetting and efficiently learning from limited novel samples, comprehensively enhancing incremental deepfake detection. We conduct extensive experiments on four popular deepfake detection datasets, which shows that our method surpasses existing incremental deepfake detection methods across four datasets. The contributions of this paper can be summarized as follows:

- We present the novel DMP model, which expands prototypes dynamically during incremental learning to adapt to new forgery patterns from limited samples for incremental deepfake detection.

- We introduce a Prototype-Guided Replay strategy coupled with a Prototype Representation Distillation loss to maintain the learned forgery patterns based on the prototypical representation of samples, thereby effectively mitigating the catastrophic forgetting in the proposed DMP model.

- We conduct extensive experiments to demonstrate the effectiveness of our method. The experimental results show that the proposed method surpasses existing incremental deepfake detection methods on multiple datasets.

## 2 RELATED WORKS

### 2.1 Deepfake Detection

The potential misuse of deepfakes presents significant risks to society, making deepfake detection an urgent yet challenging task. Early methods usually leveraged biological artifacts, such as lack of eye blinking [24], abnormal head pose [25], and atypical pupils [26] for detecting deepfake media. However, advancements in deep generative models have enabled deepfakes to avoid these obvious artifacts, rendering these methods ineffective. Subsequent works have shifted towards using end-to-end learning with handcrafted modules to detect deepfakes. For example, TALL [27] aggregated four frames and input them simultaneously into an image classifier for cross-frame comparison. AltFreezing [28] enhanced the learning of both temporal and spatial artifacts by alternately freezing the spatial and temporal modules during training. StyleGRU [29] represented the dynamic properties of style latent vectors to spot temporal distinctiveness artifacts in deepfake videos. These methods merely fit the forgery patterns within the training set, and thus their effectiveness on unseen deepfakes is limited. Another direction involves employing pretext tasks to learn audio-lip consistency representations capable of identifying deepfakes. Notable examples include lip-reading [12] and audio-visual contrastive learning [16], which are effective only for discriminating talking face videos. In general, these deepfake detectors rely on detecting specific forgery

patterns that may not exist in novel deepfakes, which leads to a decline in effectiveness. Therefore, incremental deepfake detection emerges as a more practical approach to discriminate deepfakes by directly adapting detectors to novel samples.

## 2.2 Incremental Deepfake Detection

The objective of incremental deepfake detection is to detect emerging deepfakes, which requires detectors to adapt to novel deepfakes while retaining learned forgery patterns to discriminate previously learned deepfakes. Specifically, the training procedure consists of a **base** training stage to obtain a base detector and multiple **incremental** training stages to simulate the detector's adaptation to novel deepfakes. Some related studies [30, 31] assumed the incremental stages have ample training samples. Nonetheless, as the algorithms used to generate online deepfakes are unknown, it is hard to synthesize a large volume of samples for adaptation. In practical scenarios, often only a few labeled samples are available, typically obtained through multimedia retrieval[19] or proactive deepfake watermarking [20, 21]. Therefore, in this paper, we assume only limited labeled deepfakes are available in incremental training stages. Existing few-shot incremental deepfake detection works [17, 18] emphasized retaining the learned knowledge through distillation and replay. In contrast, our method enhances the learning of novel deepfakes by modeling new deepfake patterns explicitly through prototypes and thus is more effective against emerging deepfakes.

## 3 PROBLEM DEFINITION

In incremental deepfake detection, an ideal detector is expected to continually identify new deepfakes by sequentially learning from novel samples while preserving the ability to spot previously learned deepfakes. Especially, considering the scarcity of available novel deepfakes, the detector is expected to learn novel patterns from limited samples. The continuous set of training data is formally given as $\{S^t\}_1^N$, where $S^t$ denotes the training data at $t$-th stage, and $N$ represents the total number of stages. Within the training data $S^t$ for each stage, $\{x, y\} \in S^t$ represents the corresponding training image and label, where $y \in \{0, 1\}$ denotes real or fake category, respectively. The detector is trained across different stages sequentially to learn the forgery patterns associated with the training deepfakes at each stage. We designate the first training stage ($t$=1) as the base stage and subsequent stages as incremental stages ($t$ >1). In real-world scenarios, a detector is initially trained with ample data in the base stage, which represents the pre-deployment training phase. While in preceding incremental stages corresponding to the adaptation after deployment, available deepfakes with labels are limited. Thus, the quantity of training data in the base stage $|S^1|$ is larger than those in the incremental stages $|S^t|_{t>1}$. After training in each stage, the detector is evaluated using test sets that contain samples from the current and previous stages. This evaluation setting assesses both a detector's ability to adapt to unseen deepfakes and its ability to retain learned patterns.

## 4 METHOD

We aim to enhance incremental deepfake detection, where we focus on effectively adapting detectors to limited novel deepfakes while retaining the knowledge of learned deepfakes. To adapt detectors to novel deepfakes, we introduce the Dynamic Mixed-Prototype (DMP) model, which represents a face as a Mixture of Prototypes (MoP) and adapts to novel deepfakes by expanding prototypes. To further retain learned knowledge in DMP, we propose Prototype-Guided Replay (PGR) and Prototype Representation Distillation (PRD), which are efficient in avoiding forgetting in DMP through replaying seen samples and adopting the MoP as intermediate representations for DMP to distill itself. This section is arranged as follows. We first formulate the proposed DMP model for incremental deepfake detection and detail the design motivations. Building upon the DMP model, we elaborate on the training pipeline for incrementally training the DMP model on novel deepfakes as illustrated in Figure 2. Within this framework, we describe the PGR for data replay and PRD for knowledge distillation, which aid in retaining the model's memory of previously learned deepfake representations.

## 4.1 Dynamic Mixed-Prototype Model

Learning intrinsic deepfake patterns from limited labeled deepfakes is inherently challenging, as the model is prone to overfit to biased forgery patterns when sufficient samples are unavailable. We note that the prototypical network [22] can effectively facilitate few-shot learning by projecting samples into a metric space, where compact features are learned and represented as prototypes. Since the classification within a prototypical network is conducted by calculating the distances to the prototype representations of each class, prototypical networks reflect a simpler inductive bias to reduce overfitting, which is beneficial for incremental learning.

Motivated by the advantages of prototypical networks, we introduce the Dynamic Mixed-Prototype (DMP) model to classify faces based on the distance from their embeddings to real and fake prototypes. Namely, if a face is close to a prototype in the embedding space, it contains the corresponding pattern. Previous prototype-based deepfake detection works typically adopted a single prototype for both classes to enforce a tight cluster for all deepfakes [32, 33]. We argue that each deepfake sample poses various patterns inherently, as various manipulation techniques can potentially be used to produce a single deepfake. In this circumstance, attributing all deepfakes to one fake prototype hinders the learning of richer forgery patterns. Therefore, unlike previous works, DMP adopts multiple prototypes for real or fake classes and apportions a face to multiple prototypes, *i.e.*, Mixture of Prototypes (MoP). This mixture apportionment helps to mine both shared and unique patterns in deepfakes, thereby facilitating the learning of richer patterns. Furthermore, representing faces as MoP enables increasing prototypes to expand the representations for a category. Specifically, DMP incrementally learns novel deepfakes by incorporating new prototypes for novel forgery patterns in each stage while retaining previous prototypes to preserve learned knowledge. As for real faces, the prototype set remains fixed since the distribution of real samples is consistent across different learning stages.

Formally, the proposed DMP model is built with an encoder $f$ and prototypes $P$. We denote the encoder that has been trained up to $t$-th stage as $f^t$ and the prototypes as $P^t$. $P^t$ is a set of vectors, *i.e.*, $P^t \in R^{N^t \times d}$, where $N^t$ is the number of prototypes in stage $t$ and

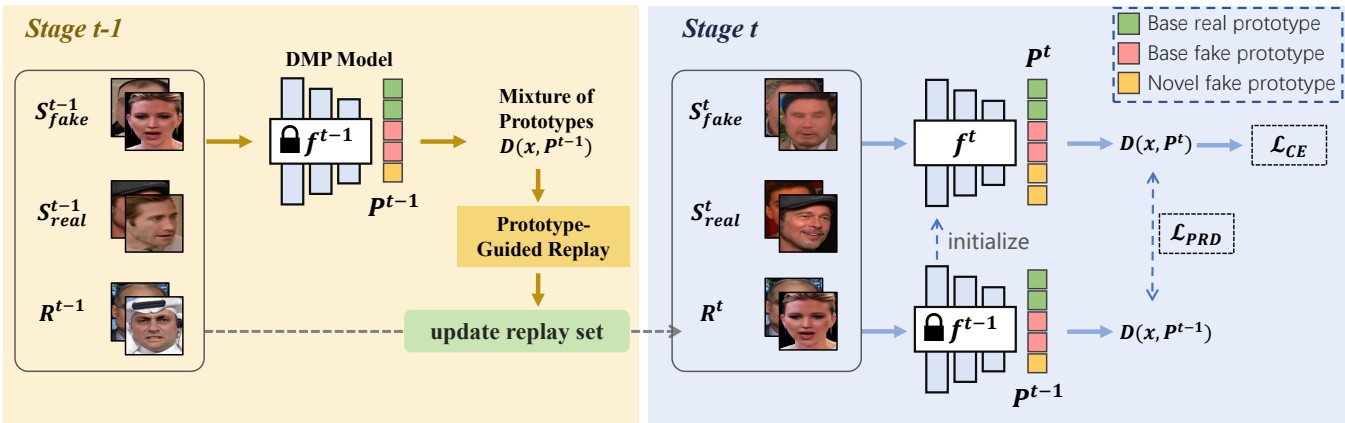

**Figure 2: The pipeline of training the proposed Dynamic Mixed-Prototype(DMP) model in stage $t > 1$. The proposed DMP consists of an encoder $f$ and a prototype layer $P$ which increases in each stage $t$. In stage $t$, the encoder from the previous stage $f^{t-1}$ serves as the teacher network and $f^t$ is distilled to preserve learned knowledge through $\mathcal{L}_{PRD}$.**

$d$ is the embedding dimension. We denote the first $k$ prototypes of $P^t$ as $P_0^t$, i.e., prototypes for real patterns, and the rest of prototypes as $P_1^t$, i.e., prototypes for forgery patterns. Initially, $P^0$ contains $k$ real prototypes. Before DMP is trained in each stage, $k$ incremental prototypes are added to $P^t$ to model the novel forgery patterns in the training data, i.e., $N^t = (t + 1) \times k$. In DMP, a face $x$ is first fed to encoder $f$ into feature embedding $f(x)$. As we aim to represent $x$ as MoP based on its similarity to $P$, its embedding is apportioned to different prototypes by calculating the distances from $f(x)$ to $P$. Specifically, we consider the softmax of the negative distances from $f(x)$ to each prototype in $P$ as the MoP, denoted as $D(x, P)$. Its $i$-th entry $D(x, P_i)$ is formalized as:

$$D(x, P_i) = \frac{\exp(-d(P_i, f(x))/\tau)}{\sum_{P_j \in P^t} \exp(-d(P_j, f(x))/\tau)}, \quad (1)$$

where $d$ is the cosine distance and $\tau = 1$ is the softmax temperature. Herein, the negative distances can be considered as similarities, and the softmax operation in its entirety transforms similarities of $f(x)$ to $P$ into normalized apportionment weights, i.e., MoP. The DMP further predicts the category of $x$ as class $c$ based on the weighted apportionment, i.e., entries in MoP $D(x, P_i)$, and the predicted classification probability $p(y = c \mid x)$ is formalized as:

$$p(y = c \mid x) = \sum_{P_i \in P_c^t} D(x, P_i), \quad (2)$$

where $c = 0$ is for real and $c = 1$ is for fake category. In training, we use cross-entropy loss $\mathcal{L}_{CE}$ for optimizing the classification:

$$\mathcal{L}_{CE} = -y \log p(y = 1 \mid x) - (1 - y) \log p(y = 0 \mid x). \quad (3)$$

During the optimization process, the feature representation $f(x)$ of a sample is attracted towards the prototypes of its corresponding category $P_i \in P_c^t$ while being repelled from the prototypes of the opposite category. After training, DMP can represent $x$ as MoP $D(x, P)$ based on its distance to prototypes, where the components of different prototypes within a sample are expressed as $D(x, P_i)$.

Prototypical networks that rely on a single prototype per class typically establish prototypes as the centroid of sample embeddings within identical classes, which ensures that the intra-class distances of embeddings are compact. However, this approach is not feasible when representing classes using multiple prototypes, as it causes overlapping among different prototypes within the same category. To circumvent this limitation, we employ pre-assigned and fixed prototypes following Yang et al. [34] and further expand their method to the scenario where one category is represented using multiple prototypes. Therefore, training DMP involves aligning the encoder $f$ with these fixed prototypes for classification. This approach prevents the collapse issue that arises when assigning prototypes as the centroid of sample embeddings, where all prototypes within the same category collapse to a single point. Additionally, the pre-assigned prototypes are largely separated. This separation encourages the encoder to learn more diverse forgery patterns, enhancing the richness of the learned deepfake representations.

## 4.2 Prototype-Guided Replay

We introduce the Prototype-Guided Replay (PGR) strategy, which is designed to select replay samples for the proposed DMP model to mitigate forgetting by stabilizing the learned deepfake representations. Specifically, PGR selects a subset of previously learned deepfake samples to be stored in a replay set, which is then incorporated into the training data for the subsequent stage. The procedure is shown in the left part of Figure 2.

Since the DMP model represents a sample as a Mixture of Prototypes, the key to maintaining the stability of representations lies in preserving the stability of prototypes. When applying PGR to select samples for replay, the closest $n$ samples to each prototype are selected at the end of each training phase. The reasons for replaying the closest samples to each prototype are twofold. First, by replaying the closest $n$ samples to each prototype, PGR helps to maintain the representational stability of the prototypes in subsequent incremental training stages. Second, since DMP uses pre-assigned and

fixed prototypes, the encoder $f^t$ is expected to map a replayed sample $x$ close to the prototype where it was initially embedded. The PGR strategy, along with the subsequent distillation loss, aids in guiding the encoder $f^t$ to project replay samples near their original prototypes. Consequently, the classification accuracy on previously learned samples of the DMP model is maintained.

The selection of the replay set is conducted at the end of each training stage. The selected replay set after stage $t$ is denoted as $R^{t+1}$. The pool of candidate samples for the replay set is the union of the current replay set $R^t$ and the current stage's training set $S^t$. For each prototype, $n$ nearest samples are selected from $R^t \cup S^t$ for replay, which are denoted as:

$$R^{t+1} = \bigcup_{P_i^t \in P^t} kNN\left(P_i^t, f^t(x^t)\right), \qquad (4)$$

where $x^t \in R^t \cup S^t$. PGR contributes to retaining the representation associated with a prototype by replaying the samples nearest to it, thus mitigating catastrophic forgetting.

## 4.3 Prototype Representation Distillation

To further mitigate catastrophic forgetting, we leverage $f^{t-1}$ to serve as the teacher model for $f^t$ to retain the knowledge learned in preceding deepfakes for training stage $t > 2$. Since DMP represents a face $x$ as a MoP $D(x, P)$ and conducts classification accordingly, we directly use $D(x, P)$ to serve as an intermediate representation from the teacher network $f^{t-1}$ to guide the training of the student network $f^t$. Specifically, we propose the Prototype Representation Distillation (RPD) loss $\mathcal{L}_{PRD}$ to align the MoP representation $D(x, P^t)$ in stage $t$ to $D(x, P^{t-1})$ in stage $t - 1$:

$$\mathcal{L}_{PRD} = KL\left(D(x, P^t) \| D(x, P^{t-1})\right), \qquad (5)$$

where $KL$ is the KL divergence. As the expansion of $P$ makes the dimensions of $D(x, P^{t-1})$ and $D(x, P^t)$ differs, we truncate $D(x, P^t)$ to match the size of $D(x, P^{t-1})$. The total loss $\mathcal{L}_{total}$ for optimizing DMP in stage $t > 1$ is:

$$\mathcal{L}_{total} = \mathcal{L}_{CE} + \alpha \times \mathcal{L}_{PRD}. \qquad (6)$$

We set $\alpha$ as 0.2 to make $\mathcal{L}_{CE}$ and $\mathcal{L}_{PRD}$ converge to the same order of magnitude.

## 5 EXPERIMENTS

### 5.1 Experimental Setup

**Dataset.** We conduct evaluations on four widely used deepfake datasets, *i.e.*, FaceForensics++ (FF++) [35], DFDC preview (DFDCp) [36] , DFD [35], and Celeb-DF v2 (CDF) [37]. **FF++** is a deepfake dataset containing 4,000 forged videos and 1,000 real videos. All videos are provided in three compression levels: raw, high-quality (HQ), and low-quality (LQ). **DFDCp** is the preview dataset for the Deepfake Detection Challenge and contains over 4,000 face swap videos. Its fake videos are generated using two face swap algorithms. **DFD** is Google's supplement to the FF++ dataset and contains over 3,000 face swap videos. **CDF** contains 590 real and 5,639 fake videos corresponding to 59 celebrities using an improved face swap algorithm. We follow the splits in [18] for training and testing.

**Evaluation protocol.** We follow the evaluation protocol in [18]. Specifically, the training sequence used in evaluation is $D_C = \{$FF++, DFDCP, DFD, CDF$\}$. Evaluated models are trained sequentially across four training sets of these stages. Upon completion of each training stage, the models are tested on the test sets of the datasets corresponding to all previously trained stages. In the base stage, the whole training set of FF++(HQ) is used for training. In incremental stages, 25 fake videos along with 25 real videos from the current incremental dataset, *i.e.*, DFDCP, DFD, and CDF, are randomly sampled for training.

**Evaluation metric.** We adopt four metrics in evaluating the proposed method following [18], *i.e.*, Accuracy, Average Accuracy, Average Forgetting, and Area Under Curve.

(1) Accuracy (ACC) refers to the ratios of correctly predicted samples in all samples. It is calculated as ACC = $\frac{TP+TN}{n}$, where $TP$ (True Positives) refers to the number of samples that are correctly identified as deepfake, $TN$ (True Negatives) refers to the number of samples that are correctly identified as real, and $n$ represents the total number of images or samples that are evaluated.

(2) Average Accuracy (AA) refers to the average accuracy of all previous stages. It is calculated as AA = $\frac{1}{N}\sum_{i=1}^{N}$ ACC$_i$, where ACC$_i$ is the accuracy of the $i$-th task and $N$ is the number of tasks.

(3) Average Forgetting (AF) refers to the average forgetting rate of previous stages. It is calculated as AF = $\frac{1}{N}\sum_{i=1}^{N}($ACC$_i^{first} -$ ACC$_i^{last})$ where ACC$_i^{first}$ and ACC$_i^{last}$ denotes the accuracy on the $i$-th task after training in stage i and current stage.

(4) Area Under Curve (AUC) refers to the area under the Receiver Operating Characteristic (ROC) curve. We use AUC to evaluate the generalization of detectors across different deepfake datasets.

**Implementation Details.** We use an Xception model pretrained on ImageNet [38] to initialize the DMP model. Our framework is trained for 40 epochs in the base stage and incrementally trained for 40 epochs in each incremental stage. The DMP model is trained using AdamW [39] optimizer with a learning rate of 2e-4 and a batch size of 64, and the learning rate is decayed to half every 5 epochs. Each batch consists of 32 real faces and 32 deepfake faces. The replay set size K is set to 500, and PGR selects $500//N_t$ nearest samples for each prototype after stage $t$ for replay. In the PyTorch framework, training with automatic mixed precision requires approximately 20GB of GPU memory.

**Preprocess.** We extract 20 frames at equal temporal intervals from each video in training and 10 frames from each video in testing. Then, the faces are aligned and cropped using Retinaface [40], maintaining a 15% margin around each face. Then, all faces are resized to $299 \times 299$ as the inputs.

### 5.2 Evaluation on Incremental Deepfake Detection

We compare the proposed method with two general incremental learning methods, *i.e.*, LWF [42] and DGR [41], and two incremental deepfake detection methods, *i.e.*, DFIL [18] and CoReD [17]. As all compared methods are model-agnostic, we use Xception as the backbone in all methods for fair comparison.

The results are shown in Table 1. Clearly, all compared methods obtained satisfying ACC on FF++ after initially training on ample data from FF++. However, when they were later incrementally

**Table 1: Performance comparison on incremental deepfake detection. †indicates reproduced using our evaluation protocol. Bold indicates top results.**

| Method | Dataset | Test Set ACC(%)↑ | | | | AA (%)↑ | AF (%)↓ |
|---|---|---|---|---|---|---|---|
| | | FF++ | DFDCP | DFD | CDF | | |
| DRG [41] | FF++ | 88.86 | - | - | - | 88.86 | - |
| | DFDCP | 78.81 | 83.89 | - | - | 81.35 | 10.05 |
| | DFD | 64.31 | 73.31 | 89.69 | - | 75.57 | 17.56 |
| | CDF | 67.33 | 79.65 | 78.35 | 76.50 | 75.45 | 12.37 |
| LWF [42] | FF++ | 95.52 | - | - | - | 95.52 | - |
| | DFDCP | 87.83 | 81.57 | - | - | 84.70 | 7.69 |
| | DFD | 76.16 | 41.78 | 96.36 | - | 71.43 | 19.89 |
| | CDF | 67.34 | 67.43 | 84.05 | 87.90 | 76.68 | 14.44 |
| CoReD [17]† | FF++ | 95.50 | - | - | - | 95.50 | - |
| | DFDCP | 92.94 | 87.61 | - | - | 90.28 | 2.56 |
| | DFD | 86.84 | 81.07 | 95.22 | - | 87.71 | 7.60 |
| | CDF | 74.08 | 76.59 | 93.41 | 80.78 | 81.22 | 11.42 |
| DFIL [18] | FF++ | 95.67 | - | - | - | 95.67 | - |
| | DFDCP | 93.15 | 88.87 | - | - | 91.01 | **2.52** |
| | DFD | 90.30 | 85.42 | 94.67 | - | 90.03 | 4.41 |
| | CDF | 86.28 | 79.53 | 92.36 | 83.81 | 85.49 | 7.01 |
| DMP (Ours) | FF++ | 95.96 | - | - | - | **95.96** | - |
| | DFDCP | 92.71 | 89.72 | - | - | **91.22** | 3.25 |
| | DFD | 92.64 | 86.09 | 94.84 | - | **91.19** | 3.48 |
| | CDF | 91.61 | 84.86 | 91.81 | 91.67 | **89.99** | 4.08 |

trained with limited data from DFDCP, DFD, and CDF, their performance in detecting previously learned deepfakes deteriorated significantly. For instance, after training on the DFDCP dataset in the second training stage, the LWF method experienced a 7.69% decrease in detection accuracy for the FF++ dataset. After the training across all four stages, its ACC scores on the FF++ and DFDCP datasets are 67.34% and 67.43%, respectively. Namely, after incremental learning with new deepfake samples, LWF failed to accurately detect previously learned deepfakes. In stark contrast, the performance of DMP to learned deepfakes consistently maintained high ACC on all subsequent datasets. For instance, after being trained up to CDF, DMP achieved an ACC of 91.61% on FF++, outperforming the top result in compared methods. Moreover, after the final stage of incremental training on CDF, DMP achieved an Accuracy Forgetting (AF) rate of 4.08%. Compared to DFIL, the proposed DMP exhibited a lower AF with identical replay set sizes. This validates the efficiency of the proposed PRD and PGR in preserving the learned forgery representations.

Besides, we note the proposed DMP is efficient in learning limited novel deepfakes. After training on the DFDCP and CDF datasets, our DMP achieved ACC scores of 89.72% and 91.67% on corresponding test sets, surpassing the highest ACC among all compared methods 88.87% (DFIL) and 87.90% (LWF). Furthermore, upon completing the training across all four stages, DMP attained an average accuracy (AA) of 89.99%, significantly outperforming all compared methods. These results validate the proposed DMP model is efficient in learning from limited novel deepfakes.

**Table 2: Comparison of the generalization ability of different methods to DFDCP, DFD, and CDF. The metric is the AUC score. The upper shows the results of cross-dataset generalization, while the lower shows the results of incremental learning. Compared results are cited from [18]. Bold indicates top results.**

| Method | Test Set AUC(%)↑ | | |
|---|---|---|---|
| | DFDCP | DFD | CDF |
| Xception[35] | 72.20 | 70.50 | 65.50 |
| LTW[43] | 74.58 | 88.56 | 77.17 |
| LRL[44] | 76.53 | 89.24 | 78.26 |
| DCL[45] | - | 91.66 | 82.30 |
| ICT[46] | - | 84.13 | 85.71 |
| UIA-ViT[15] | 75.80 | 94.68 | 82.41 |
| CoReD[17]† | 84.85 | 89.81 | 89.76 |
| DFIL[18] | 91.73 | **97.56** | 89.68 |
| DMP (Ours) | **92.37** | 97.41 | **92.84** |

**Table 3: Ablation study of the proposed model, replay strategy, and distillation method. We use the AA and AF of the detectors trained up to the last stage (CDF) for comparison.**

| Model | Replay | Distillation | AA(%)↑ | AF(%)↓ |
|---|---|---|---|---|
| Xception | [47] | [47] | 88.86 | 4.91 |
| MP | PGR | PRD | 88.45 | 5.14 |
| DMP | [47] | PRD | 88.90 | 5.21 |
| DMP | PGR | [47] | 89.10 | 5.18 |
| DMP | PGR | PRD | **89.99** | **4.08** |

## 5.3 Evaluation on Generalized Deepfake Detection

We compare the proposed DMP with recent deepfake detectors in terms of generalization using AUC as the metric. All detectors are trained on FF++ and tested on DFDCP, DFD, and CDF. The compared methods includes generalizable deepfake detection methods [15, 35, 43–46] and incremental deepfake detection methods [17, 18]. We evaluate the cross-dataset generalization performance of generalizable deepfake detection methods and the performance of incremental deepfake detection methods. The results are listed in Table 2. It is evident that generalizable deepfake detectors performed inferiorly to incremental deepfake detectors on all three datasets, which is attributed to distribution shifts across datasets. Incremental deepfake detectors exhibited higher AUC scores by adapting to 25 videos from each dataset. This suggests that when a limited number of novel deepfake samples are available for adaptation, incremental deepfake detection is a more practical approach. Notably, our DMP demonstrates the best performance among the incremental methods on the two challenging datasets, DFDCP and CDF. Our model's AUC score surpasses the best among the comparative methods on the CDF dataset by 3.08%(92.84% *v.s.*89.76%). This indicates that our DMP model can effectively learn new forgery patterns that differ significantly from learned patterns through newly added prototypes.

**Table 4: Comparison of Euclidean and Cosine distance metrics for classification in DMP.**

| Distance | Dataset | Test Set ACC(%)↑ | | | | AA (%)↑ | AF (%)↓ |
|---|---|---|---|---|---|---|---|
| | | FF++ | DFDCP | DFD | CDF | | |
| Euclidean | FF++ | 95.20 | - | - | - | 95.20 | - |
| | DFDCP | 93.35 | 87.15 | - | - | 90.25 | **1.85** |
| | DFD | 92.55 | 85.74 | 89.06 | - | 89.12 | **2.03** |
| | CDF | 91.28 | 86.50 | 85.44 | 86.27 | 87.37 | **2.73** |
| Cosine | FF++ | 95.96 | - | - | - | **95.96** | - |
| | DFDCP | 92.71 | 89.72 | - | - | **91.22** | 3.25 |
| | DFD | 92.64 | 86.09 | 94.84 | - | **91.19** | 3.48 |
| | CDF | 91.61 | 84.86 | 91.81 | 91.67 | **89.99** | 4.08 |

## 5.4 Ablation Study

**Modules.** We conduct ablations to verify the effectiveness of the proposed DMP model, PGR, and PRD. For the DMP model, we use the following variants for comparison: (1) Vanilla Xception model with a fully connected layer as the classifier instead of prototypes. As it lacks prototypes, the proposed PGR and PRD are not applicable. We instead adopt a standard exemplar sample replay and distillation method following [47]. (2) A Mixed-Prototype model similar to the DMP model with a set of fake prototypes that is not incremental, which is denoted as **MP**. For the PGR strategy, we adopt the commonly used examplar sample replay proposed in [47] for comparison. For the PRD loss, we adopt the vanilla distillation loss in [47] for comparison. The results are presented in Table 3, which indicates each element contributes to improving performance. Compared to vanilla Xception, DMP significantly enhances the AF, proving its advantages in retaining learned knowledge in prototypes. Compared to MP, DMP demonstrates a higher AA, which is attributed to the effectiveness of learning novel forgery patterns through new prototypes. Furthermore, our PRD and PGR better mitigate forgetting during incremental training compared to the replay and distillation method in [47] (fourth and fifth rows in Table 3). This is due to the fact that representing a deepfake sample as a MoP better aligns with the shared and unique features presented among different deepfake samples, thereby enhancing the effectiveness of replay and distillation.

**Distance Metric.** The DMP model classifies samples based on the distance from the samples to the prototypes. The used distance metric $d$ specifies the modeling assumption about the class-conditional data distribution in the embedding space, making the selection of the distance function vital. Common distance metrics used in prototype learning include the Euclidean distance and cosine distance, and we employ the cosine distance in DMP. We compare it with the Euclidean distance to investigate their impacts on the performance of incremental deepfake detection. The results are presented in Table 4. Employing the cosine distance as the metric achieves higher AA across all four stages, proving to be more effective than using the Euclidean distance. The Euclidean distance corresponds to the assumption of a mixed density distribution, which indicates that the distribution of sample embeddings around deepfake patterns does not align closely with the mixed density distribution. We infer that the commonly used cosine distance performs effectively due to the use of fixed prototypes. In this context, the cosine distance

**Table 5: Ablation studies for the number of prototypes parametered by $k$.**

| $k$ | Dataset | Test Set ACC(%)↑ | | | | AA (%)↑ | AF (%)↓ |
|---|---|---|---|---|---|---|---|
| | | FF++ | DFDCP | DFD | CDF | | |
| 1 | FF++ | 95.25 | - | - | - | 95.25 | - |
| | DFDCP | 91.46 | 84.97 | - | - | 88.22 | 3.79 |
| | DFD | 91.20 | 80.29 | 84.56 | - | 85.35 | 4.37 |
| | CDF | 88.68 | 81.28 | 85.51 | 88.02 | 85.87 | **3.10** |
| 2 | FF++ | 95.82 | - | - | - | 95.82 | - |
| | DFDCP | 91.87 | 87.91 | - | - | 90.39 | 3.95 |
| | DFD | 91.26 | 85.33 | 92.08 | - | 89.56 | 3.57 |
| | CDF | 90.88 | 83.49 | 90.04 | 89.63 | 88.51 | 3.80 |
| 4 | FF++ | 95.96 | - | - | - | **95.96** | - |
| | DFDCP | 92.71 | 89.72 | - | - | **91.22** | **3.25** |
| | DFD | 92.64 | 86.09 | 94.84 | - | **91.19** | **3.48** |
| | CDF | 91.61 | 84.86 | 91.81 | 91.67 | **89.99** | 4.08 |

for prototype-level classification is equivalent to a frozen linear layer, which reflects a simpler inductive bias to reduce overfitting in learning limited samples. This comparative experiment underscores the validity of our prototype layer design in the DMP model for learning limited novel samples.

**Numbers of Prototypes.** The proposed DMP model adopts multiple prototypes to represent real and fake classes. The number of prototypes added before each stage, i.e., $k$, is hand-tuned as 4, where the number of prototypes $N^t = (t + 1) \times k$. The parameter $k$ influences the DMP's ability to learn from new samples and retain knowledge about previously learned samples. This is because $k$ affects how the DMP model represents the samples as MoPs, which are then used for classification in Equation 2, replay in Equation 4, and distillation in Equation 5. We conducted ablations to assess the impact of $k$ on the performance of DMP and to verify the effectiveness of representing classes by multiple prototypes for incremental deepfake detection. The experimental results are presented in Table 5. It is obvious that when $k = 1$, the ACC in the base phase is comparable to that of using multiple prototypes $k > 1$. However, the performance of the model with $k = 1$ deteriorates notably on stage $t > 1$, whereas increasing $k$ improves the AA. Beyond the results in Table 5, we empirically found that further increasing $k$ brings marginal performance gain.

## 5.5 Sensitivity Analysis

In real-world scenarios, the size of the replay set can vary due to limitations in storage and computational resources. Moreover, the volume of accessible labeled novel deepfake samples may also change. Since the size of the replay set and the number of novel samples both impact the performance of incremental deepfake detection, we conducted sensitivity analyses on these factors separately by decreasing them. The results are shown in Figure 4. It is apparent that decreasing the size of the replay set results in a growth in AF and a decrease in AA due to the reduced accuracy on previously encountered deepfakes. We also observed that diminishing the replay set reduces the efficiency of adaptation to novel deepfakes, which further decreases the AA. This is attributed to the role of distillation, which not only mitigates forgetting but

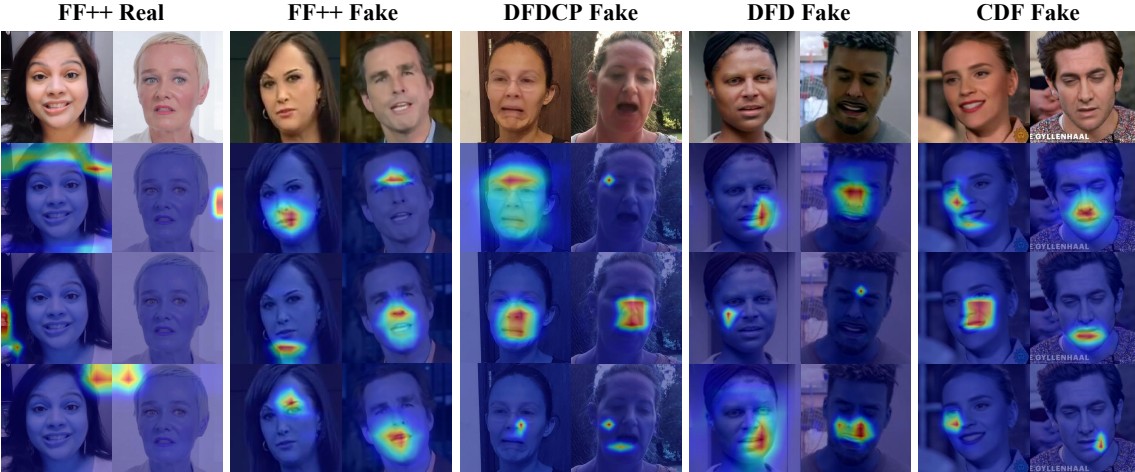

**Figure 3: Visualization of saliency areas corresponding to different fake prototypes. The first row contains real and fake faces from different training stages. The remaining three rows contain visualization results of saliency area of the three closest fake prototypes to their face embeddings in the feature space.**

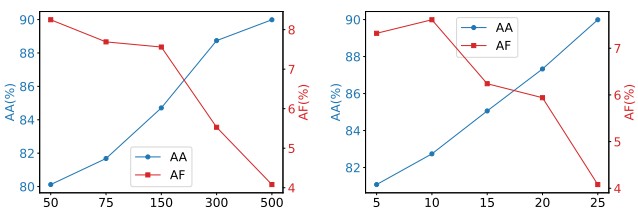

**Figure 4: Sensitivity analysis of performance AA and AF to changes in the size of the replay set (Left) and the volume of incremental deepfake training videos (Right).**

also facilitates the learning of novel deepfakes, underscoring the significance of replay and distillation for the adaptation to novel deepfakes. Besides, reducing the number of novel samples used for training also leads to a decrease in AA. The reduction in AF is less pronounced as the initial performance on novel samples is already lower due to the scarcity of available training samples and the ACC on previous test sets in subsequent training stages may improve, thereby compensating the forgetting ratio.

### 5.6 Visualization

**Saliency Map** As stated before, DMP represents each deepfake sample as a mixture of prototypes. Namely, different prototypes are expected to exhibit distinct saliency areas in a deepfake when discriminating it. We employ GradCAM [48] to visualize the component prototypes for real and fake samples to investigate their roles in discriminating deepfakes. For each sample, we visualize the corresponding saliency area of the three closest fake prototypes to its embedding in the feature space. The visualization results are shown in Figure 3. It's obvious that different prototypes correspond

to different forged areas in a deepfake sample, indicating each prototype represents a unique deepfake pattern. For instance, in the second column of samples from FF++ fake, the visualized prototypes' saliency areas respectively spot the distorted areas between the eyebrows (second row), the nose (third row), and the mouth (fourth row). Meanwhile, the combination of salient areas from different prototypes within a deepfake sample comprehensively marks major visible artifacts. This further confirms that modeling deepfakes as mixtures of prototypes aids in enriching learned deepfake representations. Besides, the saliency area visualizations for samples in FF++ real confirm that our fake prototypes avoid locating forged regions in real faces, demonstrating the efficiency of the fake prototypes in distinguishing between real and fake faces.

### 6 CONCLUSION

The proposed framework in this paper enhances the incremental learning of limited novel deepfakes while avoiding forgetting previously learned deepfakes. The proposed DMP model represents faces as mixtures of prototypes to learn rich forgery representations and thus can explicitly model new deepfake patterns with new prototypes. Moreover, to prevent DMP from forgetting learned deepfake representations, we propose PGR and PRD tailored for DMP to replay representative samples and distill to maintain the stability of existing prototypes. Extensive evaluations verify our framework can learn incremental novel deepfake samples and avoid catastrophic forgetting efficiently. In scenarios where limited novel deepfake samples are available for adaptation, our framework demonstrates satisfactory performance, which is crucial for maintaining the effectiveness of deepfake detectors. Future work includes incorporating the latest deepfake technology, such as diffusion models, to build a more comprehensive evaluation protocol for incremental deepfake detection.

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
