# OpenReview forum: "Dynamic Mixed-Prototype Model for Incremental Deepfake Detection"
_acmmm.org/ACMMM/2024/Conference — MM2024 Poster_

### Official Review · Reviewer_Yaxd · 2024-05-25

**Rating:** 4
**Confidence:** 3

**Summary:**

This paper proposed a prototype-based model for incremental deepfake detection. The authors used multiple prototypes to represent one class. In addition to the traditional CE loss, the authors introduced a prototype-based knowledge distillation loss and they also used the prototypes to guide the replay set selection.

**Strengths:**

1. The proposed method is technically sound and achieves good results on four benchmark datasets.
2. The authors performed various ablation studies to verify the effectiveness of their proposed components, such as the prototype-based replay set selection strategy and the impact of the prototype number.
3. The paper is generally well-written and easy to follow.

**Limitations:**

1. Prototype-based models have been widely used in image classification for semi-supervised and incremental learning. The idea of prototype-based knowledge distillation is not new, so the novelty of this paper is limited to some extent. The authors applied existing techniques to a new application, i.e. incremental deepfake detection which is basically an image classification problem, with moderate modifications such as the replay set selection.
2. The results reported in Table 1 are generally promising. However, I noticed that compared to DFIL, the performance gain is small on FF++, DFDCP, and DFD, but more obvious on CDF. What is the reason that the proposed method works better on the last incremental learning stage? More insightful discussions are required to explain the advantages and limitations of the proposed method.
3. It is not very clear to me how to update the prototypes at each stage. After increasing the number of prototypes by K, do you only generate K new prototypes while keeping the past prototypes unchanged, or perform clustering from scratch to update all the prototypes?

**Suitability:**

2

---

### Official Review · Reviewer_zABH · 2024-05-27

**Rating:** 4
**Confidence:** 2

**Summary:**

The Dynamic Mixed-Prototype (DMP) model addresses the challenge of detecting novel deepfakes by dynamically increasing prototypes to efficiently adapt to emerging forgery patterns. By incorporating Prototype-Guided Replay and Prototype Representation Distillation, it surpasses existing incremental detectors, demonstrating superior generalizability to novel deepfakes across multiple datasets.

**Strengths:**

The paper presents the novel DMP model, which expands prototypes dynamically during incremental learning to adapt to new forgery patterns from limited samples for incremental deepfake detection.

The paper introduces a Prototype-Guided Replay strategy coupled with a Prototype Representation Distillation loss to maintain the learned forgery patterns based on the prototypical representation of samples.

The paper conducts extensive experiments to demonstrate the effectiveness of the method.

**Limitations:**

1.  The paper emphasizes the use of cosine distance over Euclidean distance for classifying prototypes. However, this reliance might limit the flexibility of the model. If future deepfake patterns deviate significantly from the current distributions, it might be interesting to explore such experiments.

2. The DMP model, with its Prototype-Guided Replay (PGR) and Prototype Representation Distillation (PRD), involves maintaining and replaying a set of representative samples. As the diversity and volume of deepfake types grow, the storage and computational overhead for managing these prototypes and replay sets might become significant. Can be shown experimental results of some computation complexity?

**Suitability:**

2

---

### Official Review · Reviewer_XVXh · 2024-05-29

**Rating:** 3
**Confidence:** 3

**Summary:**

The paper is about a Dynamic Mixed-Prototype (DMP) Model for Incremental Deepfake
Detection. It addresses the challenge posed by the rapid advancement of deepfake technology,
which threatens social trust by creating highly convincing and realistic facial media that can be
used for malicious purposes. The paper introduces the DMP model, which dynamically increases
prototypes to adapt to novel deepfakes efficiently. This model uses multiple prototypes to
represent both real and fake classes, allowing it to learn new patterns by expanding prototypes
while retaining knowledge from previous prototypes.
The authors also propose two strategies to prevent forgetting learned knowledge: the Prototype-
Guided Replay (PGR) strategy and the Prototype Representation Distillation (PRD) loss.
These strategies are particularly effective for maintaining learned representations based on the
prototypical representation of samples.
The paper demonstrates that the DMP model, combined with the PGR strategy and PRD loss,
outperforms existing incremental deepfake detectors across four datasets and shows superior
generalizability to novel deepfakes through learning with limited deepfake samples.

**Strengths:**

The paper presents several strengths in its approach to incremental deepfake detection, which can
be highlighted as follows:
1. Novelty: The paper introduces a novel model that dynamically adapts to new deepfake
patterns by increasing prototypes. This approach is innovative as it moves beyond static
prototype representations to accommodate the evolving nature of deepfakes.
Prototype-Guided Replay (PGR) Strategy: This strategy is designed to replay samples that
are closest to the prototypes, which aids in stabilizing the learned representations and
mitigating catastrophic forgetting.
2. Theoretical Approach and Technical Correctness: The paper is theoretically sound, as it
builds upon the concept of prototypical networks, which are known for their effectiveness in
few-shot learning scenarios. The use of multiple prototypes for each class allows the model
to capture a richer set of forgery patterns.
The Prototype Representation Distillation (PRD) loss is a technically rigorous contribution
that aligns the model's representations across training stages, ensuring the stability of the
learned deepfake patterns.
3. Clarity: The paper is well-structured, with a clear introduction, detailed explanation of the
method, and a systematic presentation of the experimental results. This clarity aids in
understanding the contributions and the significance of the proposed model.
The use of figures, such as Figure 1, which contrasts the DMP model's approach with
previous methods, effectively illustrates the advantages of the proposed approach.

**Limitations:**

The paper introduces an incremental learning approach that aims to enhance the learning process
of existing detection models by incorporating a small number of new forged samples, thereby
effectively improving the model's detection generalization performance. However, there are
several issues that need further clarification in the paper:
1. It can be seen from Figure 4 that the curve does not appear to have reached its peak,
indicating that the current experimental setup may not fully reveal the potential of the
model. The ablation study presented in the figure does not adequately demonstrate the
impact of the additional samples on the model's performance. To more comprehensively
assess the model's performance, I suggest that the authors consider increasing the size of the
replay set and the number of training videos. Such improvements may help the curve to
reach a more pronounced local maximum, thereby providing a clearer performance
evaluation for the readers.
2. I have noticed that Table 2 compares the incremental learning model with existing
generalizable deepfake detection methods. To enhance the academic value and timeliness of
the paper, I suggest that the authors include at least one highly cited research work published
after 2023 in the comparison.
3. This paper presents an innovative Dynamic Mixed-Prototype model, aimed at enhancing the
application effects of incremental learning. To fully demonstrate the advantages of
incremental learning, it is suggested that the authors first present the generalization
performance of the model on cross-datasets before applying incremental learning in the
experimental section. Subsequently, the cross-dataset performance of the model on the same
datasets after incremental learning should be shown and compared. This will help to prove
that even with only a small amount of new forged method supervision data, the model can
significantly improve the performance of existing detection models.
4. In the incremental learning phase of the paper, the authors have employed a random
selection strategy to create an incremental training dataset consisting of 25 real videos and 25 fake videos. However, the impact of this random selection strategy on the stability of
detection performance is not adequately discussed in the paper. To provide a more
comprehensive assessment of the model's generalization capabilities and stability, it is
recommended that the authors conduct multiple validation experiments and report the mean
and variance of the results. This will help elucidate the effectiveness of improving the
model's generalization performance through the random selection of novel deepfake
supervised data in practical applications.
5. The DMP model has practical applications in scenarios where deepfake detectors need to be
continuously updated to handle new forgery techniques. Its incremental learning capability
makes it suitable for real-world deployment where access to novel deepfake samples might
be limited.

**Suitability:**

2

---

### Official Review · Reviewer_XfLh · 2024-06-11

**Rating:** 4
**Confidence:** 3

**Summary:**

This paper titled "Dynamic Mixed-Prototype Model for Incremental Deepfake Detection" introduces the Dynamic Mixed-Prototype (DMP) model, which aims to address the challenges posed by the rapid advancement of deepfake technology. The paper highlights that while existing deepfake detectors show promising results on familiar deepfake types, their performance significantly degrades on novel deepfakes created by unseen algorithms. The DMP model dynamically increases prototypes to adapt to novel deepfakes efficiently. It employs multiple prototypes to represent both real and fake classes, enabling the learning of novel patterns by expanding prototypes and retaining knowledge from previous prototypes. The paper also proposes the Prototype-Guided Replay (PGR) strategy and Prototype Representation Distillation (PRD) loss to prevent forgetting learned knowledge. Extensive experiments demonstrate that the DMP model surpasses existing incremental deepfake detectors across multiple datasets, showing superior generalizability to novel deepfakes.

**Strengths:**

Novelty: The introduction of the Dynamic Mixed-Prototype (DMP) model is innovative. It addresses the limitations of existing deepfake detectors by dynamically adapting to novel deepfakes through the expansion of prototypes.
Theoretical Approach: The paper provides a robust theoretical foundation for the DMP model. The use of multiple prototypes to represent real and fake classes is a significant advancement over traditional methods.
Technical Correctness: The implementation of the Prototype-Guided Replay (PGR) strategy and Prototype Representation Distillation (PRD) loss is technically sound and well-justified.
Adequate Evaluation: The paper includes extensive experiments across four datasets, demonstrating the model's effectiveness and superior performance compared to existing methods.
Clarity: The paper is well-written and clearly explains the proposed methods, theoretical background, and experimental results.
Applications: The DMP model has practical applications in enhancing the robustness and generalizability of deepfake detection systems, which is crucial for maintaining public trust and safety.

**Limitations:**

Complexity: The introduction of multiple prototypes and the additional strategies (PGR and PRD) add complexity to the model, which may affect its scalability and ease of implementation.
Computational Overhead: The dynamic expansion of prototypes and replay strategies might introduce additional computational overhead, which could be a concern for real-time applications.

**Suitability:**

2

---

### Meta-Review · Area_Chair_B5cC · 2024-06-29

**Recommendation:** Accept (Poster)
**Confidence:** 4

**Metareview:**

The paper presents the Dynamic Mixed-Prototype (DMP) Model for Incremental Deepfake Detection, designed to efficiently adapt to evolving deepfake technology by dynamically increasing prototypes to distinguish between real and fake facial media. The model utilizes multiple prototypes for each class and incorporates the Prototype-Guided Replay (PGR) strategy and Prototype Representation Distillation (PRD) loss to prevent knowledge forgetting. Testing across four datasets showed that the DMP model outperforms existing incremental deepfake detectors and demonstrates superior generalizability with limited deepfake samples.